# Embryo Development, Seed Germination, and the Kind of Dormancy of *Ginkgo biloba* L.

**Jing Feng [1,2]**, **Yongbao Shen [2,3,4,*]**, **Fenghou Shi [2,3,4]** and **Chengzhong Li [5]**

[1]  College of Landscape Architecture, Nanjing Forestry University, Nanjing 210037, China; fengjing9520@yeah.net
[2]  Collaborative Innovation Center of Sustainable Forestry in Southern China, Nanjing Forestry University, Nanjing 210037, China; fhshi406@163.com
[3]  Southern Tree Seed Inspection Center, National Forestry Administration, Nanjing 210037, China
[4]  College of Forestry, Nanjing Forestry University, Nanjing 210037, China
[5]  Department of Horticulture, Jiangsu Agri-animal Husbandry Vocational College, Taizhou 225300, China; chengzhongli11@126.com
*  Correspondence: ybshen@njfu.edu.cn; Tel.: +86-25-8542-7403; Fax: +86-25-8542-7402

**Abstract:** The embryos of *Ginkgo biloba* L. are generally reported to undergo after-ripening and be underdeveloped at the time of dispersal, which suggests that the seeds have morphological dormancy (MD) or morphological physiological dormancy (MPD). The aim of this work is to determine whether embryos of a *G. biloba* population are well-developed at the time of seed dispersal, and whether the seeds are dormant or not. From 8 September, which was the 140th day after flowering (140 DAF), seeds were collected separately from trees (T) and the ground (G) every 10 days until 7 December (230 DAF), resulting in a total of 10 samples. The changes in vertical diameter, transverse diameter, fresh weight, water content, and embryo length during seed development were measured. Simultaneously, the effects of different temperatures (15, 25, 30, and 35 °C) on seed germination, dormancy, and germination characteristics of *G. biloba* seeds were studied. Results showed that the embryos of *G. biloba* seeds were well developed and had no morphological dormancy. On 18 September (150 DAF), embryos were visible with a length of 2.5 mm. On 7 December (230 DAF), at the time of seed dispersal, their length was 17.1 mm. The germination percentage of the isolated embryos and seeds increased as the delay in seed collection increased, but there was no significant difference between T and G ($p > 0.05$). On 7 December (230 DAF), the germination of the isolated embryos reached 98%, indicating that the embryos were nondormant. Without pretreatment, seed germination was 82.57% within four weeks at 25 °C. Furthermore, the germination test at different temperatures showed the highest germination percentage at 30 °C (84.82%). Obviously, the *G. biloba* seeds were nondormant. The mean germination time (MGT) of the seeds at 30 and 35 °C was significantly lower than that of the seeds at 15 and 25 °C, and the speed of germination (SG) was significantly higher than that of the seeds at 15 and 25 °C. Although there was no significant difference in the seed-germination percentage between 30 and 35 °C, a portion of the seeds (9.5%) rotted at 35 °C. Therefore, 30 °C was the most favorable germination temperature for *G. biloba* seeds. This is the first study that reports *G. biloba* seeds with no dormancy.

**Keywords:** after-ripening; morphological dormancy; embryo development; shedding; germination; temperature; gymnosperm

## 1. Introduction

Both spore and seed plants undergo growth and reproduction stages throughout their lives. The whole process comprising the sequential, regular cycle of these two stages is called the life history or life cycle. Seeds are the sexual reproduction organ or transmission unit in higher plants, a product of the long-term evolution of plants, and a developmental stage of an individual seed plant [1]. The life history of seeds begins with flowering and pollination, then proceeds through fertilization, development, maturation, scattering, dormancy, and, finally, germination. This series of steps translate to a delicate, unique, and inter-related life course [2]. *Ginkgo* pollen matures in late April and begins to pollinate; however, fertilization does not occur immediately. The time between pollination and fertilization is approximately 120 days. At the beginning of September, the color of the ginkgo's sarcotesta changes from green to yellow or orange and white powder appears, signifying the maturity of *Ginkgo biloba* seeds [3]. Usually, the harvest period starts in about late September or earlier, but the embryo is still underdeveloped at this time. After one month, the sarcotesta begins to shrink and then a few seeds are shed, and many researchers collect their seeds during this period. For example, Cao and Cai [4] reported that the maturation and harvest period of *Ginkgo biloba* L. seeds was 20 September. Men's [5] research found that embryos grew slowly after the seeds were harvested in September. After removing the fleshy sarcotesta and placing it in a warm environment, the embryos grow to their full size (10–13 cm) within 8–10 weeks and begin to sprout [6,7]. Embryos in some seeds are differentiated at maturation and dispersal, with a radicle, a hypocotyl, and cotyledons, yet they are underdeveloped and must grow before the radicle emerges. Most of the space inside these relatively large seeds is occupied by the endosperm, with the embryos accounting for only 1% or less of the seed volume. The underdeveloped state of the embryos in these seeds means that they must grow to a "critical" size before germination. This growth causes delayed germination, which is called morphological dormancy [8]. It is generally believed that, when *G. biloba* seeds disperse, embryonic development is incomplete, the seeds are small (but differentiated), and that some time is required for the seeds to grow to a size sufficient for germination. Many seeds use dormancy to avoid adverse circumstances, and germinate only when the living environment becomes suitable [9]. Dormancy classification is based on the embryonic development state at the time of seed dispersal, the physical properties of the seeds, and the physiological responses of the seeds to environmental stimulation [10]. Nikolaeva [8] found that the type of seed dormancy was determined by the seed's morphological and physiological traits and, based on this, a seed-dormancy classification system was proposed by Baskin and Baskin [9], who divided seed-dormancy types into: physiological dormancy (PD), morphological dormancy (MD), morphophysiological dormancy (MPD), physical dormancy (PY), and joint dormancy (PY + PD). MD is hypothesized to be the oldest type of dormancy, occurring in ancient seeds (gymnosperms). The delay in seed germination may be due to a delay in fertilization, and delayed embryo development may be a response to specific light–dark, moisture, and temperature conditions, indicating that the environment plays a certain role in regulating germination time [10,11].

Even after many studies over a long time, the dormancy type of ginkgo seeds is still not clearly defined. Li and Chen [12] were the first authors to demonstrate that the underdeveloped state of the embryos at the time of dispersal was the cause of germination delay. After the seed disperses, the embryo continues to develop until cold weather comes. West et al. [6] believed that the seeds of *Ginkgo biloba* were characterized by MPD because the germination rate was only 10% after 12 weeks when they performed germination tests at room temperature ($22 \pm 2$ °C) on unstratified seeds with underdeveloped embryos, while the germination rate could be significantly increased by cold stratification or spraying exogenous $GA_3$ (90% and 75%, respectively). Wang [13] believed that ginkgo seeds were unfertilized at the time of dispersal and that fertilization occurred when the sarcotesta rotted, the germ cells divided, and the sperm was released. Cao et al. [4,14] reported that ginkgo seeds underwent after-ripening and had embryonic dormancy. The embryo was still in the prophase stage of development after morphological maturity, and germination could be achieved after a period of

maturation. The research of Qin et al. [15] suggested that low-temperature storage delayed embryo development in *G. biloba*, mainly affecting the growth of embryo length.

However, none of these authors reported any direct embryo observations or measurements to check whether they were fully developed when still on the tree. It is not known when *G. biloba* seeds naturally mature. In the existing literature, the seed dormancy of *Ginkgo biloba* and the development of the *G. biloba* embryo are still controversial. Moreover, to our knowledge, there are no detailed studies to confirm when seeds naturally disperse and whether or not the embryos undergo physiological after-ripening. Therefore, the main purpose of this study is to reveal (1) whether the seeds of *G. biloba* are dormant by observing the whole process from seed development to natural maturation and shedding, (2) the relationship between embryo size and germination, and (3) the characteristics of in vitro embryo and seed germination.

## 2. Materials and Methods

Ginkgo seeds (*Ginkgo biloba* 'Jiafozhi') were collected from three trees in Jiangsu Agri-Animal Husbandry Vocational College, Taizhou City, Jiangsu Province (32°27′38.43″ N 119°56′15.64″ E). From 8 September 2017, just when the *Ginkgo biloba* seeds reached morphological maturity (the color of the sarcotesta changed from green to orange), 50 seeds from each of the three trees (T) were randomly collected from the southeast and northwest with a tree pruner, and all fallen seeds were counted and taken from the ground to avoid mixing with the next sample (number of seeds (G) used for the germination test were the same as T). After putting them in an ice box, the seeds were immediately transported to the laboratory of Nanjing Forestry University. Thus, the first sample was taken. A total of 10 samples were taken every 10 days until the seeds were dispersed.

### 2.1. Morphological Characteristics of Seeds and Variation of Embryo Length

Three samples of 30 seeds were randomly chosen. The vertical and horizontal diameter of the seeds were measured by a vernier caliper, and the seeds were weighed with an electronic balance (SECURA513-1CN, Sartorius, Gottingen, Germany). After removing the sarcotesta and the seed coats with a pair of pliers, the seeds were cut longitudinally with a razor blade, and the lengths of the embryos were measured by stereomicroscope (SZX16, OLYMPUS Co., Tokyo, Japan).

### 2.2. Determination of Moisture Content (MC)

According to the International Seed Testing Association [16], an empty aluminum box with a lid (W1) was weighed, and the seeds were sliced into small pieces by a single-edge razor blade. Then, the samples were placed into the aluminum box (together with the lid, W2) and weighed. After drying at $130 \pm 5$ °C for 4 h in an oven (101A-1E, Shanghai Laboratory Instrument Works Co., Ltd., Shanghai, China), the samples and the aluminum box (together with the lid, W3) were weighed again. Three replications of 5 seeds were used to determine the seed MC with the following equation:

$$\mathrm{MC} = \frac{\mathrm{W2} - \mathrm{W3}}{\mathrm{W2} - \mathrm{W1}} \times 100\% \tag{1}$$

where W1 is the weight of the container with the lid; W2 is the weight of the container with the lid and the sample before drying; and W3 is the weight of the container with the lid and the sample after drying.

### 2.3. Seed Germination Characteristics

#### 2.3.1. In Vitro Embryo Culture

Three samples of 50 embryos were excised from the seeds and incubated on absorbent cotton moistened with deionized water in plastic germination boxes; then, they were incubated in a growth chamber at a constant temperature of 25 °C. The embryos were observed in vitro every day and

germination was recorded when the radicle had elongated to 2 mm over the 14 day incubation period. The absorbent cotton was kept moist during the whole process, and the germination rate of the in vitro embryos was determined after the end of the culture period.

2.3.2. Germination Test

Four samples of 50 seeds were taken every 10 days from 7 November (200 days after flowering (DAF)) to 17 December (230 DAF). According to the International Seed Testing Association [16], the seeds were put on absorbent cotton moistened with deionized water in plastic germination boxes and incubated in a growth chamber at a constant temperature of 25 °C with an 8 h photoperiod. Seed germination was recorded when the radicle had elongated to 2 mm (n). Seed germination was scored every 3 days over the 30 day incubation period. *G. biloba* seeds experience an inembryonate phenomenon, so to eliminate the effect of embryoless seeds (NE) on the tests and obtain an accurate germination percentage, both the germination percentage and embryoless seeds were accounted for.

The total germination (TGP) [17] was calculated by the following formula:

$$\text{TGP} = \frac{n}{N - NE} \times 100\% \tag{2}$$

where TGP is the seed-germination percentage; n is the number of seeds that germinated normally; N is the sample size of the tested seeds (50); and NE is the number of embryoless seeds.

The mean germination time (MGT) [18] and the speed of germination (SG) [19] were calculated simultaneously.

$$\text{MGT} = \frac{\sum(D{\cdot}n)}{\sum n} \tag{3}$$

where MGT is the mean germination time; n is the number of seeds which germinated on day D; and D is the number of days counted from the beginning of germination.

$$\text{SG} = \sum\left(\frac{n}{t}\right) \tag{4}$$

where SG is the speed of germination; n is the number of seeds newly germinating at time t; and t is the day from sowing.

2.3.3. Effect of Temperature on Seed Germination

On 7 December, four temperature gradients (15, 25, 30, and 35 °C) were set up for the germination of seeds that matured naturally and were shedding. Four samples of 50 seeds were randomly selected for each temperature. The other germination conditions were the same as in Section 2.3.2, and the germination percentage was calculated at the end of germination.

*2.4. Statistical Analyses*

The data were analyzed with SPSS 23.0 (IBM, Armonk, NY, USA) and Excel (Microsoft Office Home and Student Edition 2016, Microsoft Corporation, Redmond, WA, USA) software. Statistical analysis was performed using one-way or two-way analysis of variance (ANOVA) followed by Duncan's Multiple Range Test (DMRT). Values are expressed as the mean ± SD (standard deviation) of three replicates in each of the independent experiments; *p*-Values < 0.05 were considered statistically significant.

**3. Results**

*3.1. Embryo Development*

Fertilization of *Ginkgo biloba* occurred in middle and late August, and the zygote proceeded to rapidly undergo continuous mitosis, forming embryos after experiencing the free nuclear and

cellular stages [20–22]. The process of embryo development can be divided into three stages: proembryo development, embryonic differentiation, and late embryonic development. At the beginning of proembryo formation, when the original embryo was formed, there was no apparent polar differentiation in morphology, but with the division of the cells, polar differentiation began to occur in the tissue. The differentiation of *Ginkgo biloba* embryos took about a month.

As shown in Figure 1A, from 18 September (150 DAF), when the embryo was visible, with a length of 2.5 mm, the shape of the proembryo was stemlike, the cotyledon primordia were differentiated, and the top of the embryo formed an arc depression. On 28 September (160 DAF), the chalazal end cells were divided into two cotyledons, the micropylar end cells formed undeveloped suspensor tissue, and the embryo length was 4.5 mm. On 8 October (170 DAF), the cotyledon, germ, hypocotyl, and radicle were gradually differentiating, and the embryo length was 8.6 mm. Thereafter, at the late embryonic development stage, the embryo continued to grow. On 18 October (180 DAF), the chalazal end endosperm cells disintegrated to form the embryo cavity, and the cotyledons turned green. On 17 November (210 DAF), the length of the embryo had increased to 16.5 mm (Figure 2); after that, the embryo slowly grew to its size at the time of seed dispersal, 17.1 mm (7 December, 230 DAF), at which point it basically filled the whole embryo cavity. Figure 1B shows the morphological changes of the sarcotesta, of which the color was initially pale yellowish-green (140 DAF) but gradually changed to orange (160 DAF), yellowish-brown (180 DAF), and grayish-brown covered with a thick layer of white powder (230 DAF). The sarcotesta began to shrink at 140 DAF and was easily peeled off. The volume of the *G. biloba* seeds had decreased to its minimum at 230 DAF, that is, 3/5 of its size at 140 DAF—a decrease of 39%. The measurement of the vertical and transverse diameter of *G. biloba* seeds (Table 1) showed that the vertical diameter of the seeds remained fairly constant from 150 to 230 DAF, while the transverse diameter of the seeds decreased significantly ($p < 0.05$) and the water content of the seeds decreased significantly ($p < 0.05$) during the time period of 150–200 DAF. This was caused by the loss of water in the sarcotesta during the late stage of development (Figure 1B, Table 1). According to the correlation analysis of each index, as shown in Table 2, there were significant negative correlations between embryo length and seed transverse diameter, water content, and fresh weight during the development of *G. biloba* seeds (transverse diameter correlation = –0.942, $p < 0.01$; water content correlation = –0.908, $p < 0.01$; fresh weight correlation = 0.865, $p < 0.01$), indicating that there was a close relationship between embryo development and seed water content. The time of onset of embryo development was late in the *G. biloba* seeds, and the embryos were well-developed at the time of dispersal. Therefore, there was no morphological dormancy (after-ripening) in these embryos.

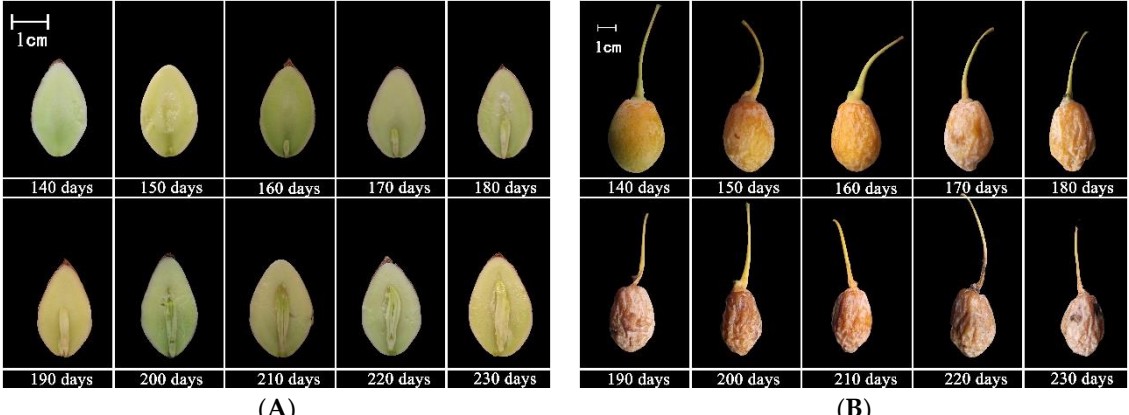

**Figure 1.** Morphological changes during embryo development in *Ginkgo biloba* L. 'Jiafozhi' Bar = 1 cm. (**A**) 140–230 days after flowering (DAF): embryo development of *Ginkgo biloba* L. 'Jiafozhi' seeds; (**B**) 140–230 DAF: the morphological changes of the sarcotesta.

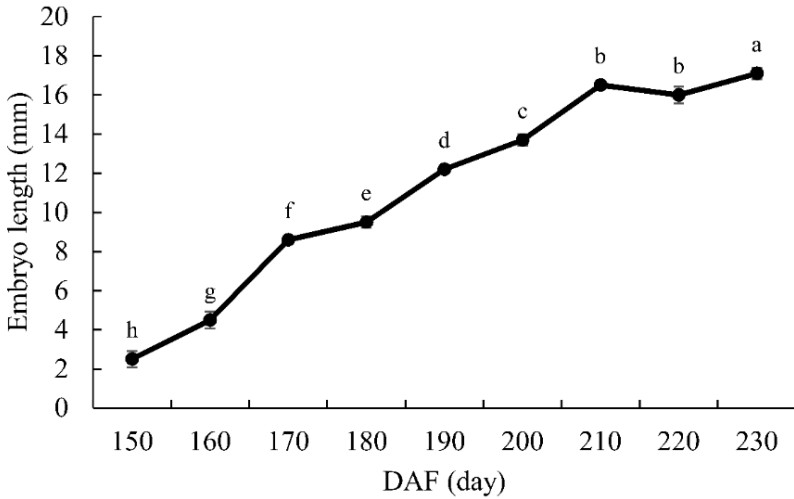

**Figure 2.** Changes in embryo length during the seed development of *Ginkgo biloba* 'Jiafozhi'. Data are the mean (±SD) of three replicates, and bars labeled with different lowercase letters are significant at *p* < 0.05.

**Table 1.** Multiple comparisons of the external morphological changes of different stages during embryo development of *Ginkgo biloba* 'Jiafozhi'.

| DAF | Date | Fresh Weight (g) | Water Content (%) | Vertical Diameter (mm) | Transverse Diameter (mm) | Embryo Length (mm) | Features |
|---|---|---|---|---|---|---|---|
| 150 | 18 September 2017 | 10.10 ± 1.00 a | 64.27 ± 0.33 ab | 31.18 ± 0.17 a | 22.86 ± 0.30 a | 2.5 ± 0.40 h | Orange-yellow, slightly wrinkled |
| 160 | 28 September 2017 | 10.09 ± 0.33 a | 65.49 ± 0.55 a | 31.35 ± 0.24 a | 22.33 ± 0.27 b | 4.5 ± 0.44 g | idem |
| 170 | 8 October 2017 | 9.17 ± 0.32 ab | 63.67 ± 0.96 b | 30.78 ± 0.92 ab | 21.83 ± 0.29 c | 8.6 ± 0.40 f | idem |
| 180 | 18 October 2017 | 8.50 ± 0.14 bcd | 61.63 ± 0.72 c | 30.87 ± 0.22 ab | 20.83 ± 0.30 d | 9.5 ± 0.11 e | Yellowish-brown, wrinkled |
| 190 | 28 October 2017 | 8.58 ± 0.85 bc | 61.46 ± 0.90 c | 30.20 ± 0.27 bc | 20.57 ± 0.16 de | 12.2 ± 0.64 d | Brown, covered with white powder, wrinkled, stem shrank |
| 200 | 7 November 2017 | 7.44 ± 0.30 cd | 58.36 ± 0.76 d | 31.51 ± 0.73 a | 20.61 ± 0.41 de | 13.7 ± 0.50 c | idem |
| 210 | 17 November 2017 | 7.38 ± 0.40 d | 58.27 ± 1.53 d | 30.86 ± 0.35 ab | 20.30 ± 0.10 e | 16.5 ± 0.61 ab | idem |
| 220 | 27 November 2017 | 7.71 ± 0.34 cd | 57.35 ± 1.03 d | 29.87 ± 0.20 c | 19.61 ± 0.30 f | 16.0 ± 0.70 b | idem |
| 230 | 7 December 2017 | 7.62 ± 1.05 cd | 56.97 ± 1.03 d | 30.13 ± 0.35 bc | 19.54 ± 0.21 f | 17.1 ± 0.66 a | Grayish-brown, covered with white powder, wrinkled, fruit stalk shrank |

Note: the same lowercase letter does not differ significantly at the 0.05 probability level.

**Table 2.** Correlation analysis between embryo length, vertical diameter, transverse diameter, water content, and fresh weight during seed development of *Ginkgo biloba* 'Jiafozhi'.

| Indicators | Embryo Length | Transverse Diameter | Vertical Diameter | Water Content | Fresh Weight |
|---|---|---|---|---|---|
| Embryo Length | 1 | −0.942 ** | −0.464 * | −0.908 ** | −0.865 ** |
| Sig. (two-tailed) | | 0.000 | 0.015 | 0.000 | 0.000 |
| Transverse Diameter | | 1 | 0.564 ** | 0.898 ** | 0.779 ** |
| Sig. (two-tailed) | | | 0.002 | 0.000 | 0.000 |
| Vertical Diameter | | | 1 | 0.423 * | 0.275 |
| Sig. (two-tailed) | | | | 0.028 | 0.165 |
| Water Content | | | | 1 | 0.809 ** |
| Sig. (two-tailed) | | | | | 0.000 |
| Fresh Weight | | | | | 1 |

* Significant (Sig.) at 0.05 $p$ level, ** significant at 0.01 $p$ level; others are nonsignificant.

### 3.2. Seed Dispersal

As shown in Figure 3, as time elapsed, the number of shed *G. biloba* seeds increased. Between 8 September (140 DAF) and 28 October (190 DAF), the total number of dropped seeds was only 388; however, the number increased dramatically during the period of 210–220 DAF: during these 10 days, it reached its highest value of 800. Analysis of variance showed that the number of shed seeds at 220 DAF was significantly higher than that at 210 DAF ($p < 0.05$). Maximum shedding for *G. biloba* was concentrated in the period of 210–230 DAF, accounting for 82.7% of the total number of shed seeds. As indicated in Table 1, embryo length was 17.1 mm, and only 20% of the seeds were embryoless. The remaining seed embryos were well-developed, which indicates that the embryos of *G. biloba* were well developed at the time of seed dispersal.

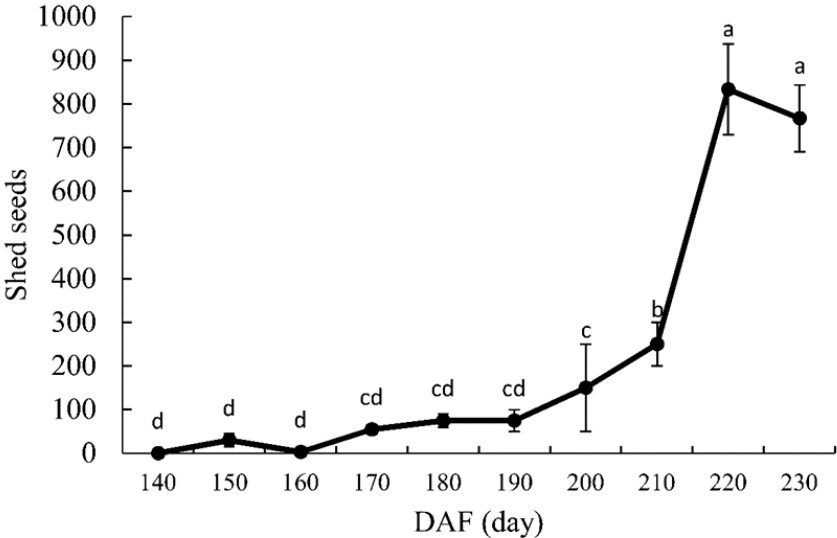

**Figure 3.** Number of seeds shed during the seed development of *Ginkgo biloba* 'Jiafozhi'. Data are the mean (±SD) of three replicates, and bars labeled with the different lowercase letters are significant at $p < 0.05$.

### 3.3. In Vitro Embryo Culture

Figure 4 shows the germination percentages of excised embryos that were collected at different stages. On 8 October (170 DAF), the germination percentages after a 14 day incubation of excised embryos of *G. biloba* seeds collected from the T and from the G were only 5.9% and 13.3%, respectively. However, on 28 October (190 DAF), the germination percentage of T and G reached 81.2% and 83.55%,

respectively. The variance analysis showed that the germination percentage of excised embryos at 190 DAF was significantly different from 170 DAF ($p < 0.05$). With longer delays in seed-collection time, the germination percentage of excised embryos continued to increase, and at 220 DAF, the germination percentages of T and G had increased to the highest values of 97.8% and 98.6%, respectively. Moreover, the germination process of excised embryos from T and G was similar, and there was no significant difference in germination percentage between T and G at any of the time points ($p > 0.05$).

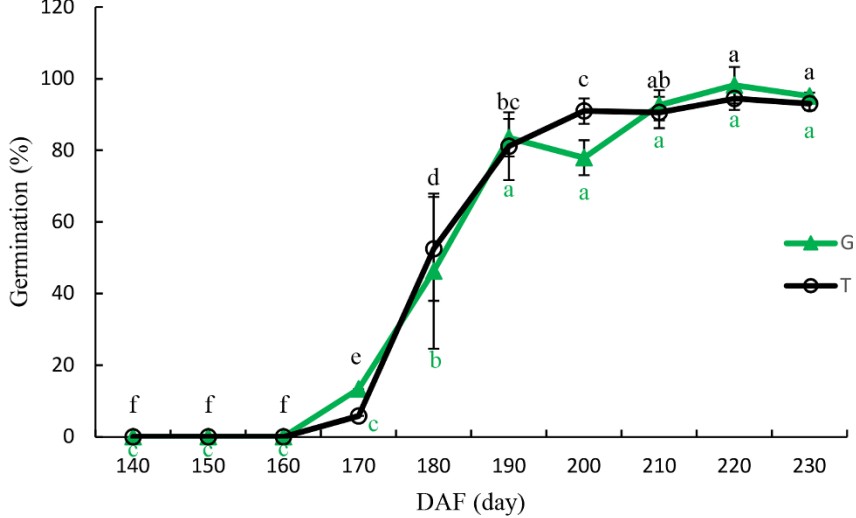

**Figure 4.** Germination of isolated embryos from trees (T) and the ground (G) of *Ginkgo biloba* 'Jiafozhi'. Data are the mean (±SD) of three replicates, and bars labeled with different lowercase letters are significant at $p < 0.05$.

### 3.4. Seed Germination

The effects of different seed-collection times on the germination of the seeds from T and the G are shown in Table 3. On 7 November (200 DAF), the germination percentages of T and G were 59.33% and 57.03%, respectively, and on 27 November (220 DAF), they were 80.68% and 71.96%, respectively. In addition, the germination percentage of T was significantly higher at 220 DAF than that at 200 and 210 DAF ($p < 0.05$), while there was no significant difference in germination between 220 and 230 DAF. At 230 DAF, the germination rate of G was significantly higher compared to the rates at 200, 210, and 220 DAF, and, as shown in Table 1 and Figure 3, the seeds matured and dispersed at this time. This indicates that seed-collection time had a significant effect on the germination of *G. biloba* seeds (Table 4), while the source of the seeds (T or G) had no significant effect on the germination, indicating that the germination percentage of *G. biloba* seeds was not related to the seed-collection method, but increased with the delay in seed-collection time.

**Table 3.** Effects of seed-collection time on the germination of seeds from T and G.

| Time (Day) | Germination (%) | |
| --- | --- | --- |
| | **T** | **G** |
| 200 DAF | 59.33 ± 8.14b | 57.03 ± 10.90b |
| 210 DAF | 55.00 ± 5.00b | 56.51 ± 10.25b |
| 220 DAF | 80.68 ± 9.03a | 71.96 ± 5.57b |
| 230 DAF | 81.30 ± 3.95a | 82.57 ± 5.32a |

The same lowercase letter does not differ significantly at the 0.05 probability level. T: Seeds from trees; G: Seeds from the ground.

**Table 4.** Effects of different seed collection time and methods on the germination of *Ginkgo biloba* 'Jiafozhi'.

| Source | Sum of Squares | df | Mean Square | F | p |
|---|---|---|---|---|---|
| Corrected Model | 3727.075 [a] | 7 | 532.439 | 9.029 | 0.000 |
| Intercept | 113,324.925 | 1 | 113,324.925 | 1921.645 | 0.000 |
| Methods | 3.150 | 1 | 3.150 | 0.053 | 0.820 |
| Time | 3535.747 | 3 | 1178.582 | 19.985 | 0.000 * |
| Methods × Time | 188.178 | 3 | 62.726 | 1.064 | 0.392 |
| Error | 943.566 | 16 | 58.973 | | |
| Total | 117,995.566 | 24 | | | |
| Corrected Total | 4670.641 | 23 | | | |

* Significant at 0.05 *p* level; others are nonsignificant. df: Degree of freedom; F: F value; [a]: R-Squared = 0.798 (Adjusted R-Squared = 0.710).

At 25 °C, the water gradually saturated the endosperm of *G. biloba* seeds and the tissue reached an imbibition state. At 10 days after sowing, the seed suture line was cracked, and a reddish-brown membranous endopleura could be seen (Figure 5A). The radicle started to sprout and broke through the micropyle and the oval protective film of the endopleura. Then, embryo tissue began to emerge from the fissure through the gap of the endopleura (Figure 5B).

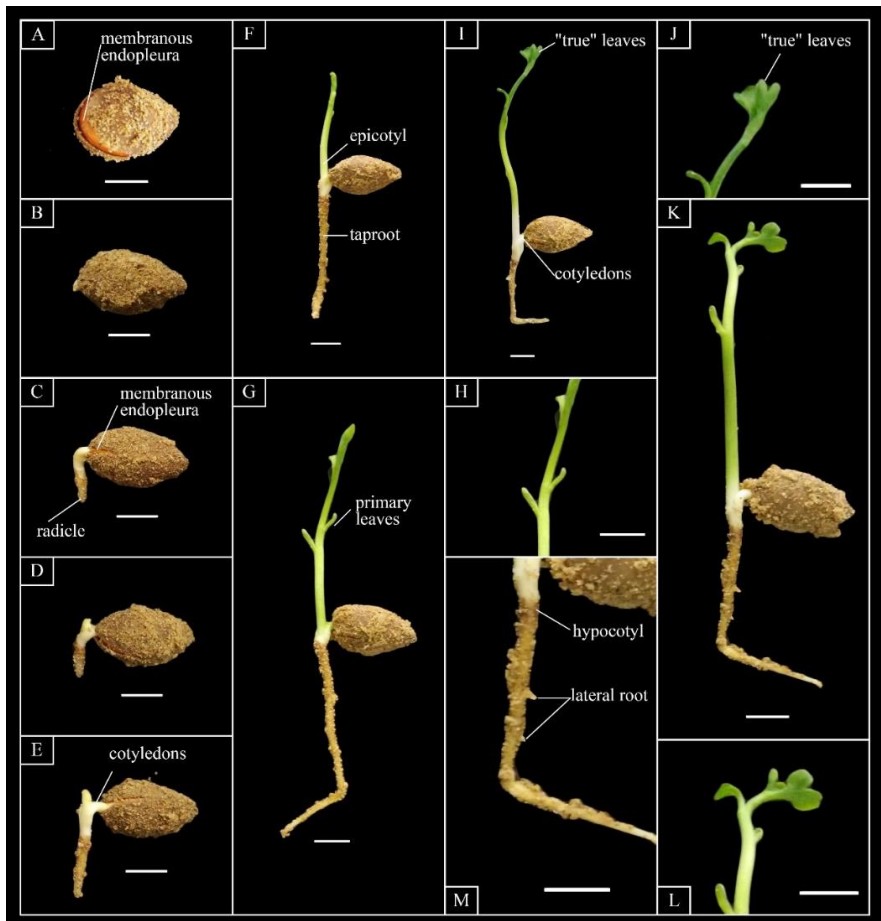

**Figure 5.** Process of germination and seedling development of *Ginkgo biloba* 'Jiafozhi'. Bar = 1 cm. (**A**) Seed dehiscent, showing reddish-brown membranous endopleura; (**B**) seed after germination; (**C**) emerged primary root; (**D**) leaf primordia-forming buds; (**E**) bud elongation; (**F**) erect branch; (**G**) primary leaves; (**H**) enlarged image of primary leaves from (**G**); (**I**) "true" leaves; (**J**) enlarged image of the "true" leaves from (**I**); (**K**) ginkgo seedling; (**L**) enlarged image of fan-shaped leaves from (**K**); (**M**): enlarged image of lateral roots from (**K**).

After germination, the division and differentiation of the embryo cells began to accelerate, and the speed of growth significantly increased. The primary root emerged after the micropylar end of the seed was ruptured (Figure 5C). While the cotyledon grew thicker, the lower part of the cotyledon became obviously elongated, and the radicle and hypocotyl continued to be pushed out of the seed coat. The base of the cotyledon was swollen and extended in a sheath shape. With the formation of the cotyledon sheath, the embryonic cells between the cotyledons divided and formed a small dome shape. When the radicle and germ developed to a certain extent, polar differentiation was obvious: the germ grew upward, and the radicle grew down. The root primordium of the radicle continuously differentiated so that it was obviously enlarged, and the apex extended into the soil and grew downward, forming the main root. The upper part of the cotyledons almost stopped elongating, and it was still contained in the endosperm and continued to supply nutrients, while the lower part of the cotyledons continued to grow and elongate until it protruded from the seed coat and appeared bent and arched (Figure 5C). Then, the germ at the base began to differentiate to form the first vegetative leaf primordium. During the immediate period after, the main root continued to grow thicker, and stretched into a cylindrical shape. The root tip slowly extended down into the soil, and the root system was initially established. When the young root grew to 0.5–0.8 cm in diameter and 0.5–1.5 cm in length, the first vegetative leaf primordium began to develop into a bud (Figure 5D,E). Then, the bud sprouted from the gap at the base of the cotyledons and stretched upward. The cotyledon was still wrapped in endosperm at the time of germination, and its lower surface was in close proximity to the endosperm; this allows the cotyledon to absorb nutrients from the endosperm, which supplies nutrition to various parts of the embryo for growth. The epicotyl and young stem were pale-green and stout. The diameter was about 0.5 mm and tapered so that it was finer toward the top. The young stem continued to grow upward, forming erect branches (Figure 5F). The first leaves were undeveloped primary leaves, which are always scale leaves, with 2–4 alternate leaves that were wide and linear (Figure 5G,H), 3–5 cm long, 2 mm wide, and flat or retuse at the apex. Leaves at the third or fifth position gradually changed to "true" leaves with the petiole elongated, and fan-shaped leaves also slowly formed (Figure 5I,J). When the "true" leaves unfolded, they were 1 cm long and 0.5 cm wide with a deep fissure at the apex. The edge of the leaf was irregularly wavy (Figure 5K,L) and the base of the leaf was wedge-shaped; the veins were dichotomies, the surface of the leaves was green, and the abaxial surface was light-green covered with powder. The length of the petiole was 0.4 cm. As the leaves matured, the texture changed from soft to hard, and the color changed from light-green to dark-green. The hypocotyl and main root of the seedlings were not hypertrophic. The hypocotyl was very short, white, and smooth, with a length of 1.5–5 mm. The diameter of the collar was about 3.5 mm, the taproot was thick and strong, the lateral roots were short (Figure 5M) and yellowish-brown, and the tip of the radicle was white.

*3.5. Effect of Temperature on Germination*

As shown in Figure 6, the seeds of *G. biloba* grown at 15 °C began to germinate on the 21st day, and the germination process was slow. On the 30th day, seed germination was only 16.59%. The germination percentage generally increased with increasing temperatures: it was 82.57% at 25 °C, and the highest germination percentage was 84.82% at a temperature of 30 °C. However, it decreased to 73.73% at 35 °C due to a large number of the seeds rotting (9.5%) during the incubation. From Table 5, compared to 15 °C, the other three temperatures had a significant effect on the germination percentage of *G. biloba* seeds ($p < 0.05$), but there were no statistically significant differences between the three temperatures. The MGT of *G. biloba* seeds at 30 °C and 35 °C was significantly lower than that at 15 °C and 25 °C, and the SG at 30 and 35 °C was significantly higher than that at 15 and 25 °C, which indicates that higher temperatures were favorable for the germination of *G. biloba* seeds. Furthermore, there was no significant difference in germination percentage, SG, and MGT between 30 and 35 °C, but the rot rate of *G. biloba* seeds at 35 °C was significantly higher than that at 30 °C. Therefore, 30 °C was the most suitable germination temperature for *G. biloba* seeds.

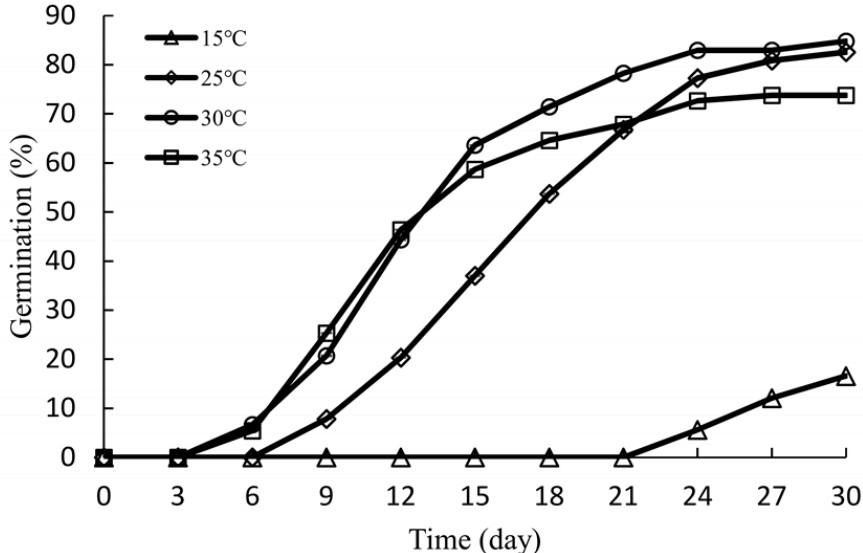

**Figure 6.** Germination of *Ginkgo biloba* 'Jiafozhi' seeds at different temperatures. Data are the means of three replicates ± SD.

**Table 5.** Effect of temperature on germination of *Ginkgo biloba* 'Jiafozhi' seeds.

| Temperature (°C) | Germination (%) | Mean Germination Time (MGT) (day) | Speed of Germination (SG) (%) | Rot Rate (%) |
|---|---|---|---|---|
| 15 | 16.59 ± 11.87 b | 26.93 ± 0.53 a | 0.27 ± 0.21 c | 0.00 ± 0.00 c |
| 25 | 82.57 ± 7.67 a | 17.57 ± 0.85 b | 2.18 ± 0.22 b | 3.00 ± 2.58 bc |
| 30 | 84.82 ± 1.69 a | 14.05 ± 1.28 c | 3.10 ± 0.47 a | 9.50 ± 4.12 b |
| 35 | 73.73 ± 9.81 a | 13.14 ± 0.42 c | 2.98 ± 0.43 a | 19.00 ± 7.39 a |

The same lowercase letter does not significantly differ at the 0.05 probability level.

## 4. Discussion

### 4.1. Morphological Changes in Seed Development of G. biloba

Studies on the embryo development of *G. biloba* have resulted in ambiguity. Some authors have reported observations of the early embryo development of *G. biloba*. For example, Ball [23] confirmed maturation of tissue was not complete in the roots of these in vitro seedlings. Wang and Chen [24] suggested coconut milk, both filtered and autoclaved, which was tested for its effect on the growth and structure of the young embryos. However, due to the long time from pollination to fertilization (108–112 days), in most studies, seed collection was carried out in the middle of September, since the volume of *G. biloba* seeds was no longer increasing and the sarcotesta was orange-yellow, soft, and slightly shrunken, thus having a mature appearance. However, no research has reported the whole process of *G. biloba* embryo development while the seeds were on trees. In this study, most of the embryos were not yet completely differentiated in mid-September. According to the anatomical structure of the seed, the embryos differentiated in early October, including cotyledons, germs, hypocotyls, and radicles. Thereafter, the embryos continued to grow rapidly and were fully developed in early December (reaching two-thirds of the seed length). Seeds were also naturally shed, indicating that there was no morphological dormancy in the embryo of *G. biloba*, and the embryos had fully developed on the tree. The phenomenon of delayed development in the embryos of *G. biloba* has caused confusion in researchers. The reason for delayed embryo development may be to prevent seeds from germinating prematurely in autumn so they can survive the cold winter [25].

### 4.2. Seed Maturation and Dispersal

The seeds of most plants shed rapidly at maturity, while other seeds take months to years to fall off, such as the seeds of *Pinus banksiana* Lamb. and *Pinus albicaulis* Engelm. [26]. Gymnosperms may control the time of seed germination by regulating the time of seed maturation and shedding, or by delaying germination until environmental conditions become favorable [10]. *G. biloba* displays the seed-dropping phenomenon throughout its whole growth and development. Early seed dropping generally lasts from early May to early July, accounting for 10% of the total, but it can reach 40%–50% in more extreme circumstances. The reasons for early seed dropping are related to the nutritional status of the tree, poor pollination, hormones, and climate. It was found that from 8 September (140 DAF) to 28 October (190 DAF), the number of falling *G. biloba* seeds accounted for only 8% of the total, while from 27 November (220 DAF) to 17 December (230 DAF), it was 82.7%. The results indicate that seeds matured and shed during this period. To our surprise, our results showed the opposite of the popular opinion that the embryos are small and need after-ripening when seeds disperse. Thus, the collection of *G. biloba* seeds for the purpose of establishing seedlings should be delayed until the time of seed dispersal.

### 4.3. Seed Germination and Isolated Embryo

Previous studies have examined the isolated embryos of *G. biloba*. For example, Wang et al. [27] reported that the embryos of *G. biloba* collected in Beijing on 24 September were 0.7–1.4 mm long. The cotyledon had not differentiated, and royal jelly could promote organ differentiation and the development of young embryos of *G. biloba*. Using an in vitro embryo culture of *G. biloba* with 0.2% glucose solution or distilled water, Li [28] found that the former was beneficial to the growth of young embryos, but, in the latter, the cotyledon and hypocotyl were obviously grown only in large embryos. The radicle was slightly prolonged, yet no further growth occurred. When analyzing the studies of these authors, we found that most of the seeds used in their experiments were collected in September and were in the early stage of embryo development, and most of the embryos were undifferentiated. In this study, the germination results of embryos isolated from *G. biloba* seeds collected at different stages showed that the germination of isolated embryos increased significantly during the 10 days between 18 October (180 DAF) and 28 October (190 DAF). This was 30% higher than the sample collected in the previous period, which is probably because most of the embryos differentiated during this period. Usually, in warm temperate regions, fertilization of *G. biloba* occurs in August, and embryos begin to develop in early October [29]. Wang et al. [30] observed the development of *G. biloba* in Beijing for three years and found that fertilization occurred during 16–20 August and, by 26 October, the embryo had differentiated, including the cotyledon, germ, hypocotyl, and radicle. The germination percentage of the isolated embryos remained at a relatively low value, indicating that the embryos had not differentiated, and most of the embryos could not germinate. However, a few embryos still could germinate, which may be due to the influences of external environmental conditions and nutrient competition between seeds. There are great differences in fertilization, even in the same plant, because the phase from unformed sperm to the proembryo could exist simultaneously in the same plant [31]. As time delays in seed collection increased, the germination percentage of isolated embryos increased. At 190 DAF, the germination percentage reached 80%, indicating that the embryos were nondormant.

Willis et al. [10] reported that, when seeds with MD were mature and shedding, the embryos had completely differentiated into the cotyledon, hypocotyl, and radicle, but they were still small and needed to grow to a critical size before germination. In related books and articles on the germination of *G. biloba* seeds, it is common to find ginkgo seeds described as dormant. Most researchers have suggested that the embryos in *G. biloba* seeds are incompletely developed and have morphological or morphophysiological dormancy. However, in this study, we found that the embryos were well-developed (reaching two-thirds of the seed length) at the time of seed dispersal. Without pretreatment, seed germination reached >80% within four weeks at 25 and 30 °C, and the

germination rate of decoated seeds reached 80% within two weeks [32]. Therefore, the *G. biloba* seeds were nondormant.

### 4.4. Effect of Temperature on Seed Germination

A suitable temperature can promote the germination of seeds and the growth of seedlings. The optimum germination temperature of different seeds may vary, and even among seeds of the same species, temperatures have different effects depending on seed origin [33]. Temperature affects seed germination by affecting enzyme activity, respiration rate, and water-absorption rate [34]. When the temperature is too low, the activation or catalysis ability of the enzyme is inhibited, whereas a high temperature leads to the inactivation of the enzyme, the destruction of the cell-membrane system and organelles, and the inhibition of the physiological metabolism of the seeds. Therefore, an unfavorable temperature is not conducive to seed germination [35]. For example, the time at which the germination percentage of *Clinopodium sandalioticum* (Lamiaceae) seeds reached 50% decreased with the increase in temperature from 10 °C (18 days) to 20 °C (6 days) [36]. Higher temperatures (27–35 °C) were suitable for the seed germination of *Cycas revolute* [37,38]. Yu et al. [39] reported that the seed germination of *G. biloba* at 25 °C was higher than that at 15 °C, and the respiration rate at 15 °C was significantly lower than that at 25 °C. In this experiment, when the temperature was 35 °C, although the initial germination time of *G. biloba* seeds was earlier than that at 30 °C, the germination percentage was lower than that at 30 °C. These results indicate that the higher temperature caused significant damage to the seeds, which succumbed to mildew and rotted during germination. On the other hand, at 15 °C, the germination process was very slow. Although the germination percentage at 25 °C was also high, the MGT and SG had a better response at 30 °C. Therefore, 30 °C was the optimum temperature for the germination of *G. biloba* seeds. This is inconsistent with the Xia et al. [40] study, where 25 °C was the reported optimum temperature for germination.

### 5. Conclusions

In conclusion, this study confirmed that: (i) The embryos of *G. biloba* seeds were well-developed at the time of seed dispersal and the embryos were nondormant. Without any stratification treatment, seeds of *G. biloba* achieved a high germination of 82.57% at 25 °C. It was clear that seeds of *G. biloba* were nondormant. (ii) High temperature (30 °C) was beneficial to the germination of *G. biloba* seeds. In conclusion, different from the existing literature reporting that ginkgo seeds were dormant, we report the first, to our knowledge, embryos of *G. biloba* seeds that were well-developed at the time of seed dispersal and embryos that were nondormant. This was mainly due to the seeds collected by researchers not being naturally matured. Moreover, without any stratification treatment, seeds achieved a high germination percentage; therefore, the seeds of *G. biloba* were nondormant. A high temperature (30 °C) was beneficial to the germination of G. biloba seeds. Nevertheless, it should be noted that we did not investigate the relationship between the endosperm and embryo during seed development, the metabolism of the endosperm, or the changes in endogenous hormone content, which are especially relevant during embryo development; this relationship still needs to be fully elucidated. Further experiments will be conducted to fully clarify these points.

**Author Contributions:** J.F., Y.S., and F.S. participated in the discussion and experimental designs; C.L. prepared the experiment materials; J.F. undertook laboratory analyses and drafted the manuscript. All authors read and approved the final manuscript.

**Funding:** This research was funded by the Priority Academic Program Development of Jiangsu Higher Education Institutions (PAPD).

**Conflicts of Interest:** The authors declare no conflict of interest.

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
