# Peer review of "Embryo Development, Seed Germination, and the Kind of Dormancy of Ginkgo biloba L."

_forests, doi:10.3390/f9110700_

Round 1
Reviewer 1 Report
Dear Authors, Great job.
Please find my specific comments in methodology especially on germination rate and speed of germination and provide accurate information and find my general comments throughout the manuscript. The title need some minor modification please find my suggestion for the title. Rest of the comment are editorial.

Author Response
Dear Reviewer:
Thank you for your comments concerning our manuscript entitled "Embryo Development, Germination Characteristics and Kind of Dormancy in Seed of Ginkgo biloba" (ID: forests-379618). Those comments are all valuable and very helpful for revising and improving our paper, as well as the important guiding significance to our researches. We have studied comments carefully and have made correction which we hope meet with approval. Revised portion are marked in red in the paper. The main corrections in the paper and the responds to the reviewer’s comments are as flowing:
RESPONSES TO COMMENTS OF REVIEWER #1
1. Line 2: The title needs some minor modification please find my suggestion for the title.
Response: Thank you for the suggestion, the title has been changed to “Embryo development, Seed Germination and Kind of Dormancy in Ginkgo biloba”. (Line 2)
2. Line 25: …and had no morphological dormancy…
Response: We have changed the “non-morphological dormancy” into “no morphological dormancy” in the revised version. (Line 26)
3. Line 56: make sure to follow the Forests journal guidlines for authors. Is that underline under ref numbers? if yes, please delete them.
Response: Thank you for the suggestion, we have deleted the underlines according to reviewer's instructions.
4. Line 108: be consistence and use upper case for heading and subheadings same as line 120, 121, and 128
Response: we have used upper case according to the Journal's Rules and checked the rest parts of the article carefully.
5. Line 137-138: Any reference for this formula? I believe this is Total germination% not Germination rate? Germination rate is closer to the meaning of SG in line 147 for example time to reach to 50% of germination or time to reach to 90% of germination consider as a germination rate.
Response: We are grateful for the suggestion and very sorry for our inaccurate expression. Indeed, the formula referred to the total seed germination percentage rather than the germination rate, which was more related to the speed of germination. To be more clearly and in accordance with the reviewer concerns, we have replaced the “The germination rate (GR)” with “The total germination (TGP)”. (Line 150)
The following reference has been added:
[17] Duclos, D.V.; Ray, D.T.; Johnson, D.J.; Taylor, A.G. Investigating seed dormancy in switchgrass (Panicum virgatum L.): understanding the physiology and mechanisms of coat-imposed seed dormancy. Industrial Crops and Products 2013, 45, 377-387.
6. Line 147: SG is n is? Please delete "is n"
Response: Thank you for the suggestion, the "is n" was deleted according to the reviewer’s suggestion. (Line 160)
7. Line 236: Suggestion: Since the paper will be publish in color how about using different line and marker colors for T and G
Response: Thank you for the suggestion, we have modified it according to the comment. (Line 252)
8. Line 410-414: Please do not repeat the results in this section and write a statement to conclude your research
Response: Thank you for the suggestion again, we have revised this section according to the comment.
The following text has been added:
In conclusion, different from the existing literature that ginkgo seeds were dormant, we report the first, to our knowledge, embryos of G. biloba seeds that were well developed at the time of seed dispersal and embryos that were nondormant. It was mainly due to the seeds collected by researchers were not naturally matured. Moreover, without any stratification treatment, seeds achieved a high germination percentage; therefore, the seeds of G. biloba were nondormant. A high temperature (30 °C) was beneficial to the germination of G. biloba seeds… (Line 442-447)
The following text has been deleted:
In conclusion, this study confirmed that: (i) The embryos of G. biloba seeds were well developed at the time of seed dispersal and the embryos were non-dormant. Without any stratification treatment, seeds of G. biloba achieved a high germination of 82.57% at 25℃. It was clearly that seeds of G. biloba were non-dormant. (ii) High temperature (30 ℃) was beneficial to the germination of G. biloba seed… (Line 438-442)
9. Line 446, 477: References, something is missing here.
Response: We would like to thank the reviewer for the comments about references. The missing space has been inserted in line 489 and we have carefully checked the other references. The reference in line 535 has been corrected to:
[38] Grobbelaar, N. Cycads, with Special Reference to the Southern African Species; Nat Grobbelaar: Pretoria, 2002.
Once again, thank you very much for your valuable advice.

Reviewer 2 Report
The manuscript shows a complete and comprehensive study on germination characteristics in Gingko biloba. It is an interesting study that provides valuable information to understand the first processes during germination in this plant.
Here are some comments for the authors’ consideration:
General editorial comment: There is a lack of gaps between the last word in sentence and brackets with a number of citation
Other comments:
Line 3: Species name „Ginkgo biloba” should not be italicized?
Line 57: The authors used the terms “radical” and “radicle” in the MS, the meaning is the same, but it is good to use one term throughout the MS
Line 70: Why “Morphological dormancy” is capitalized and other used terms of dormancy are not?
Line 85: Why “Embryonic morphology dormancy” is capitalized and other used terms of dormancy are not?
Line 102: Word “ were” is italicized, should not be.
Line 152: Sentence “The other germination conditions were the same as in 1.2.3,…”. Is there the right number provided? I could not find section 1.2.3. in the MS.
Line 189: There is a lack of gaps in “…from150 DAF” and 190DAF”.
Lines 199-201: “ 140d-230d DAF” I suggest remove “d”. DAF abbreviation means Day After Flowering, so it is a kind of repetition of information. Information about the bar is provided two times in figure caption; maybe one time is enough.
Line 208: Unnecessary “dot” before “water”.
Line 209: Species name „Ginkgo biloba” should not be italicized?
Line 226-227: Sentence is unclear and I suggest rewriting it.
Line 229: ”however” should not be capitalized?
Lines 229, 233-235: What is “S”? Should not be “S” replaced by “G”?
Line 232: Please provide a number of DAF after “December 17”
Lines 243, 246, 249: What is “S”? Should not be “S” replaced by “G”?
Lines 244-246: Sentence is unclear and I suggest rewriting it.
Line 254: To clarify data given in table 4, it would be good to provide the information what is i.e. df, F. What does mean the second asterisk used in table 4?
Lines 256-294: In the whole fragment, authors refer to the Figure2. Figure 2 shows changes of embryo length, not the germination process. I think that authors should refer to Figure 5.
Line 258: The authors use term “coleoptile”. Coleoptile is the protective sheath covering the emerging shoot in monocotyledons such as grasses. Please provide information, what kind of tissue of G. biloba embryo is a coleoptile or give a literature citation about the existence of coleoptile in G. biloba.
Lines 270,272, 278: I suggest using the term “cotyledons” instead of “dicotyledons”.
Figure 5D: The authors use term “coleorhiza”. Coleorhiza is the protective sheath covering the emerging root in monocotyledons such as grasses. Please provide information, what kind of tissue of G. biloba embryo is a coleorhiza or give a literature citation about the existence of coleorhiza in G. biloba.
Lines 297-298: The sentence “radical broke through the coleoptile” is incorrect. If G. biloba has this kind of tissues, this sentence should be “radical broke through the coleorhiza”.
Line 296: In my opinion Figure 5 shows the process of germination and seedling development of G. biloba.
Line 300: Picture F from figure 5 shows one seedling.
Line 303: Effect of temperature on germination is shown in Figure 6 and Table 5.
Author Response
Dear Reviewer:
Thank you for your comments concerning our manuscript entitled "Embryo Development, Germination Characteristics and Kind of Dormancy in Seed of Ginkgo biloba" (ID: forests-379618). We thank you for the time and effort that you have put into reviewing the previous version of the manuscript. The valuable comments not only helped us with the improvement of our manuscript, but suggested some neat ideas for future studies. We have studied comments carefully and have made correction which we hope meet with approval. Revised portion are marked in green in the paper. The main corrections in the paper and the responds to the reviewer’s comments are as flowing:
RESPONSES TO COMMENTS OF REVIEWER #2
1. General editorial comment: There is a lack of gaps between the last word in sentence and brackets with a number of citations.
Response: Thank you for the suggestion. According to the Reference Guide of the Journal, the citation number have to be close to the Author citation. We have revised this throughout the manuscript.
2. Line 3: Species name “Ginkgo biloba” should not be italicized?
Response: The Latin/scientific names “Ginkgo biloba” have been corrected as italicized. (Line 4)
3. Line 57: The authors used the terms “radical” and in the MS, the meaning is the same, but it is good to use one term throughout the MS.
Response: Thank you for the suggestion, we keep the terms “radicle” consistent throughout the manuscript.
4. Line 70: Why “Morphological dormancy” is capitalized and other used terms of dormancy are not?
Response: We have changed the “Morphological dormancy” into “morphological dormancy” in the revised version. (Line 76)
5. Line 85: Why “Embryonic morphology dormancy” is capitalized and other used terms of dormancy are not?
Response: We have changed the “Embryonic morphology dormancy” into “embryonic morphology dormancy” in the revised version. (Line 92)
6. Line 102: Word “were” is italicized, should not be.
Response: Thank you for the suggestion again, we have revised it according to the comment.
7. Line 152: Sentence “The other germination conditions were the same as in 1.2.3…”. Is there the right number provided? I could not find section 1.2.3. in the MS.
Response: We would like to say sorry to the reviewer and the editor for the careless mistake, Sentence “The other germination conditions were the same as in 1.2.3…” has been corrected to “The other germination conditions were the same as in 2.3.2…” (Line 166)
8. Line 189: There is a lack of gaps in “…from150 DAF” and 190DAF”.
Response: These missing spaces have been inserted and we have checked the other parts of text carefully.
9. Lines 199-201: “140d-230d DAF” I suggest remove “d”. DAF abbreviation means Day After Flowering, so it is a kind of repetition of information. Information about the bar is provided two times in figure caption; maybe one time is enough.
Response: Thank you for the suggestion again, “140d-230d DAF” has been replaced with “140-230 DAF” and the bar has been used one time in figure caption. (Line 215)
10. Line 208: Unnecessary “dot” before “water”.
Response: The “dot” has been deleted and we have checked the rest parts of the text carefully.
11. Line 209: Species name „Ginkgo biloba” should not be italicized?
Response: Thank you for the suggestion again, the Latin/scientific names “Ginkgo biloba” have been corrected as italicized. (Line 224)
12. Line 226-227: Sentence is unclear and I suggest rewriting it.
Response: We are grateful for the suggestion and very sorry for our inaccurate expression. We have rewritten the sentence “Figure 4 showed that the germination of excised embryos which were collected from different stages” as “Figure 4 shows the germination percentage of isolated embryos which were collected at different stages” in the revised manuscript.
13. Line 229: “however” should not be capitalized?
Response: Thank you for the suggestion again, it should be capitalized, so we have modified the “however” to “However” in the revised manuscript.
14. Lines 229, 233-235: What is “S”? Should not be “S” replaced by “G”?
Response: We would like to thank the reviewer for the comments about this issue. Indeed, we misspelled “G” as “S”, and “S” has been replaced with “G” in these lines.
15. Line 232: Please provide a number of DAF after “December 17”
Response: Thank you for the suggestion. We have added “(220 DAF)” after “December 17”.
16. Lines 243, 246, 249: What is “S”? Should not be “S” replaced by “G”?
Response: Thank you for the suggestion again. Indeed, we misspelled “G” as “S”, and “S” has been replaced with “G” in these lines.
17. Lines 244-246: Sentence is unclear and I suggest rewriting it.
Response: In Line 224-246, the sentence “In addition, the germination percentage of T was significantly higher than 200 DAF (p < 0.05), while there was no significant difference in germination between 220 DAF and 230 DAF” did exist some confusion, and it has been rewritten as: “In addition, the germination percentage of T was significantly higher at 220 DAF than that at 200 and 210 DAF (p < 0.05), while there was no significant difference in germination between 220 and 230 DAF”.
18. Line 254: To clarify data given in table 4, it would be good to provide the information what is i.e. df, F. What does mean the second asterisk used in table 4?
Response: The ambiguity might be caused by the unclear description in our manuscript, so we would like to say sorry to the reviewer. The second asterisk used in Table 4 means the interaction between the two factors (methods and time), we have replaced it with “×”
19. Lines 256-294: In the whole fragment, authors refer to the Figure2. Figure 2 shows changes of embryo length, not the germination process. I think that authors should refer to Figure 5.
Response: Thank you for underlining this deficiency. Actually, we referred to figure 5. The mistakes have been modified throughout the text one by one.
Line 258: The authors use term “coleoptile”. Coleoptile is the protective sheath covering the emerging shoot in monocotyledons such as grasses. Please provide information, what kind of tissue of G. biloba embryo is a coleoptile or give a literature citation about the existence of coleoptile in G. biloba.
Response: The ambiguity might be caused by the unclear description in our manuscript, so we would like to say sorry to the editor and the reviewer. The use of this term “coleoptile” was not correct. Indeed, there was no coleoptile in embryo of Ginkgo biloba. The basal end of the ginkgo embryo assumes the status of a meristem pressed against the venter from which the shoot apical meristem and cotyledons develop. The radicle develops from cells immediately behind this meristematic region. The micropylar end of the seed is the first to rupture and the root tip emerges from the seed coat.
And now we have corrected the text as follows:
“The radicle started to sprout, and the apical coleoptile broke through the micropyle and the oval protective film of endopleura…” This sentence has been corrected as:
“The radicle started to sprout, and broke through the micropyle and the oval protective film of the endopleura”. (Line 274-275)
20. Lines 270,272, 278: I suggest using the term “cotyledons” instead of “dicotyledons”.
Response: Thank you for the suggestion, the term “cotyledons” have been used instead of “dicotyledons”.
21. Figure 5D: The authors use term “coleorhiza”. Coleorhiza is the protective sheath covering the emerging root in monocotyledons such as grasses. Please provide information, what kind of tissue of G. biloba embryo is a coleorhiza or give a literature citation about the existence of coleorhiza in G. biloba.
Response: We are grateful for the suggestion and very sorry for our inaccurate expression. The use of this term “coleorhiza” was not correct. Indeed, there was no coleorhiza in embryo of Ginkgo biloba. To be more clearly and in accordance with the reviewer concerns, we have modified figure 5D.
22. Lines 297-298: The sentence “radical broke through the coleoptile” is incorrect. If G. biloba has this kind of tissues, this sentence should be “radical broke through the coleorhiza”.
Response: Thank you for the suggestion again, the use of some terms was not very accurate in our text, and we have revised “radical broke through the coleoptile” as “the primary root emerged”. The sentence “The apex of the radicle broke through the coleoptile and produced the root primordium” in line 262-263 has been modified as “the primary root emerged after the micropylar end of the seed was ruptured”.
23. Line 296: In my opinion Figure 5 shows the process of germination and seedling development of G. biloba.
Response: Thank you for the suggestion again, the caption of Figure 5 has been modified according to the Reviewer’s comment. (Line 314)
24. Line 300: Picture F from figure 5 shows one seedling.
Response: Thank you for underlining this deficiency. “Ginkgo seedlings” in picture F from figure 5 has been replaced by “a ginkgo seedling”. (Line 318)
25. Line 303: Effect of temperature on germination is shown in Figure 6 and Table 5.
Response: Thank you for underlining this deficiency. We are very sorry for our incorrect writing, and we have been corrected these mistakes one by one in the revised manuscript.
Thank you for suggestions, and all of your suggestions are very important for my thesis writing and research work. Thank you for your valuable advice again.

Reviewer 3 Report
Is a good paper with a focus on the embryo development, seed dormancy and germination in Ginkgo biloba seeds. This study makes a meaningful contribution to our understanding of seeds of this species and provides new data compared to previous reports.
However, the manuscript needs some revisions that should be addressed. In particular, I found many careless mistakes made by the authors (e.g. many mistakes in the text formatting, wrong figure reference number in the text (two times!), reference [16] not found cited in the text, page numbers are incorrect etc.). In addition, the English must be revised, I highly recommended to submit the manuscript to a linguistic review process. All comments and suggestions are find in the PDF document in the attachment.

Author Response
Dear Reviewer:
Thank you for your comments concerning our manuscript entitled "Embryo Development, Germination Characteristics and Kind of Dormancy in Seed of Ginkgo biloba" (ID: forests-379618). Those comments are all valuable and very helpful for revising and improving our paper, as well as the important guiding significance to our researches. We have studied comments carefully and have made correction which we hope meet with approval. Revised portion are marked in blue in the paper. The main corrections in the paper and the responds to the reviewer’s comments are as flowing:
RESPONSES TO COMMENTS OF REVIEWER #3
1. Line 2: I suggest to modify the title as "Embryo Development, Seed Germination and Kind of Dormancy of Ginkgo biloba"
Response: We are grateful for the suggestion. The title has been changed according to the Reviewer’s comment.
2. Line 17: Change with: The aim of this work was to
Response: Thank you for the suggestion. We have changed “With the aim” with “The aim of this work is to” (Line 18)
3. Change with: if (Line 18);
Response: Thank you for the suggestion again. This sentence has been corrected as: “The aim of this work is to determine whether embryos of a G. biloba population are well developed at the time of seed dispersal and whether the seeds are dormant or not.” (Line 18-19)
4. Insert missing spaces
Response: These missing spaces in lines (26, 27, 28, 55, 56, 61, 67, 71, 78, 81, 83, 87, 89, 122, 124, 129, 165, 180, 189, 200, 201, 216, 227, 231, 235, 245, 276, 284, 292, 296, 305-312, 314, 315, 326, 337, 340, 354, 359, 369, 374, 379, 385, 391, 396, 398-402, 404, 408) have been inserted and we have checked the other parts of text carefully.
5. This sentence has to be moved after the full stop in the line 32.
Response: We have moved this sentence according to the reviewer’s comment. (Line 34)
6. Delete “the; reached highest; SG and MGT; 25-; morphological physiological dormancy; the; morphological; of embryos; seed; s; immediately; at random; their; so; n is; has obvious stages, which; stage; stage; stage; ,; and; diameter; ,; of G. biloba seeds; the; ,; And the; speed of germination; mean germination time; 25-; or; of; the rate; seeds; 25-; didn’t be fully elucidated.”
Response: we have deleted these words one by one in Lines (32, 35, 36, 79-80, 83, 86, 90, 93, 104, 106, 109, 132, 135, 147, 165, 166, 167, 169, 174, 186, 187, 190, 204, 208, 230, 310, 311, 315, 351, 374, 384, 398, 406, 416).
7. add: "highest" between "the" and "germination"; "embryo" between "direct" and "observations"; "seed" between "the" and "dormancy"; "immediately" between "were" and "transported"; "randomly" between "were" and "chosen"; "statistically" between "considered" and "significant"; "of" between 20% and "seeds"; "and 210 DAF"; before "H" add "G,"; "," after seeds; "," after Usually; "30°C"; "For example" before "the"; ","
Response: we have added these words one by one in Lines (32, 90, 92, 105, 109, 159, 218, 284, 362, 365, 385, 395, 396).
8. Replace: "." with ","; "." with ","; "to" with "who"; "3" with "three"; "these" with each; "germinated" with "were incubated"; "Seed germination was"; "Seed germination was scored"; "seed germination percentage"; "the number of seeds which"; "for"; "the mean" ; "." with ","; with "the mean (± SD) of three replicates,"; ";" with "."; "The maximum shedding period"; "the mean (± SD) of three replicates,"; 220 and 230 DAF; "the mean (± SD) of three replicates,"; "compared to"; "." with ":"; "However,”; "were"; "the germination results"; replace with "germination"; "germination"; ">80%"; "may vary"; "among seeds of the same species"; "seed germination percentage"; "In addition,"; "Xia et al. [30]”; "seeds achieved a high germination , therefore seeds"; "experiments will be made"
9. Response: we have replaced these words one by one in Lines (52, 56, 62, 77, 103, 104, 123, 132, 133, 139, 144, 152, 158, 176, 204, 207, 216, 222, 235, 238, 246, 297-300, 325, 354, 361, 363, 366, 369, 384, 389, 404, 405, 407, 412, 417).
10. Line 39: I suggest to add another keyword which refers somehow to the species... for example “Gymnosperm”
Response: Thank you for the suggestion, “Gymnosperm” has been added to the keywords.
11. Please, add the reference
Response: Thank you for the suggestion again. The following reference has been added:
[1] Fenner, M.; Thompson, K. The ecology of seeds; Cambridge University Press: UK, 2005; pp. 1-29.
[2] Fenner, M. Seeds: the ecology of regeneration in plant communities., 2nd ed.; CABI Publishing: Wallingford, UK, 2002; pp. 331-361.
[3] Del Tredici, P. The phenology of sexual reproduction in Ginkgo biloba: Ecological and evolutionary implications. Bot. Rev. 2007, 73, 267-278.
[12] Li, T.T.; Chen, S. Temperature and the development of Ginkgo embryo. Sci. Rept. Nat. Tsing Hua Univ., Ser. B 1934, 2, 37-39.
[28] Li, T.T. The development of Ginkgo embryo in Vitro Sci. Rept. Nat. Tsing Hua Univ., Ser. B 1934, 7, 169-174.
The missing reference has been inserted:
[33] Ghildiyal, S.K.; Sharma, C.M. Effect of Seed Size and Temperature Treatments on Germination of Various Seed Sources of Pinus wallichiana and Pinus roxburghii from Garhwal Mmalaya. Indian For. 2005, 131, 56-65.
[35] Bewley, J.D.; Bradford, K.; Hilhorst, H.; Nonogaki, h. Seeds: Physiology of Development, Germination and Dormancy, 3rd ed.; Plenum Press: New York, NY, USA, 2014.
12. Line 54: Please, you should cite at least some examples.
Response: We are grateful for the suggestion.
The following text has been added:
… For example, Cao and Cai [4] reported that the maturation and harvest period of Ginkgo biloba seeds was September 20. Men’s [5] research found that the embryos grew slowly after the seeds were harvested in September …
The following reference has been added:
[4] Cao, B.; Cai, C. Advances in Physiology of Ginkgo Seeds. Shandong Agriculture Sciences 2001, 1, 40-42.
[5] Men, X. Structure and developmental rhythm of Ginkgo biloba seeds. Deciduous Fruits 1989, 1, 20-22.
13. Line 71: According to the Reference Guide of the Journal, the citation number have to be close to the Author citation. Please you should revise this throughout the manuscript.
Response: Thank you for underlining this deficiency. We have revised this throughout the manuscript.
14. Line 76: This cited reference lacks in the references list. Please, when you will add Li and Chen in the reference list, remember to add also the corresponding number citation here.
Response: Thank you for the suggestion again, we have added this reference according to the reviewer’s comment.
The following reference has been added:
[12] Li, T.T.; Chen, S. Temperature and the development of Ginkgo embryo. Sci. Rept. Nat. Tsing Hua Univ., Ser. B 1934, 2, 37-39.
15. Line 102: should not be in italics
Response: Thank you for the suggestion again, we have revised it according to the comment.
16. Line 104: How many seeds did you collect from the ground? the same number of T? please specify it.
Response: The ambiguity might be caused by the unclear description in our manuscript, so we would like to say sorry to the editor and the reviewer. Actually, each time sampling, we collected all the seeds that have fallen off from the three trees to avoid mixing with the next sample. The seeds (G) used for the germination test were the same number of T. We have modified it in the revised manuscript.
The following text has been added:
… all … to avoid mixing with the next sample (The number of seeds (G) used for the germination test were the same of T) …
17. Line 106: this sentence is not necessary
Response: Yes, we have deleted this sentence.
18. Line 108: You should use upper case according to the Journal's Rules
Response: Thank you for the suggestion, we have used upper case according to the Journal's Rules and checked the rest parts of the article carefully.
19. Line 110, 112, 117: please add the technical balance information. At least the Model and the readability (decimal places) ecc...
Response: Thank you for the suggestion again, the following text have been added:
… (SQP, Sartorius, Gottingen, Germany) … (Line 119)
… (SZX16, OLYMPUS Co., Tokyo, Japan) … (Line 121)
… (101A-1E, Shanghai Laboratory Instrument Works Co., Ltd., Shanghai, China) … (Line 126)
20. Line 119: Please, specify better W1, W2 and W3 in the text. As written is not clear.
W1 = weight of container with lid;
W2 = weight of container with lid and sample before drying
W3 = weight of container with lid and sample after drying
Response: Thank you for underlining this deficiency. We have specified W1, W2 and W3 according to the Reviewer’s suggestion.
The following text have been added:
… where W1 is the weight of the container with the lid; W2 is the weight of the container with the lid and the sample before drying; and W3 is the weight of the container with the lid and the sample after drying. … (Line 129-131)
21. Line 124: Please, temperatures (°C) must be coherent throughout the manuscript. When present, you have to delete the space between the number and °C (for example, 25 °C, 30 °C) in all the cited temperatures.
Response: Thank you for the suggestion. We have kept the °C symbol homogeneous throughout the manuscript (with a space between value and magnitude), and we have checked carefully and corrected the rest in the whole text.
22. Line 124: modify the sentence in this way: “were observed every day and germination was recorded when radicle”
Response: Thank you for the suggestion. We have modified the sentence according to the Reviewer’s comment. (Line 136-137)
23. Line 132: at light conditions? if yes, at which photoperiod?
Response: Yes, with an 8 h photoperiod.
The following text have been added:
… with an 8 h photoperiod … (Line 144)
24. Line 137: This formula it referes to the total seed germination percentage rather than the germination rate, which is more related to the speed germination. I suggest to replace "The germination rate" with "The total germination"
Response: We are grateful for the suggestion. We have replaced “The germination rate” with “The total germination”. (Line 150)
25. Line 152: Do you mean the paragraph 2.3.2?
Response: Yes. We would like to say sorry to the reviewer and the editor for the careless mistake, and we corrected “1.2.3” to “2.3.2”. (Line 166)
26. Line 171: modify the text in this way “the embryo was visible, with a length of 2.5 mm,”
Response: Thank you for the suggestion. We have modified the sentence according to the Reviewer’s comment. (Line 184-185)
27. Line 181: modify the sentence in this way: "the morphological changes of sarcotesta, which colour was"
Response: Thank you for the suggestion. We have modified the sentence according to the Reviewer’s comment. (Line 195)
28. Line 189: Why did you consider only the range until 190, without 200 DAF? From table 1, I noted that the water content decreased significantly also in 200 DAF compared to 150 DAF
Response: Thank you for underlining this deficiency. Indeed, the water content decreased significantly in 200 DAF compared to 150 DAF. We have replaced “190 DAF” with “200 DAF”. The sentence was rewritten as: “…and the water content of the seeds decreased significantly (p < 0.05) during the time period of 150–200 DAF.” (Line 202-203)
29. Line 190: it is not necessary repeat it, we already know that we are speaking about G. biloba seeds
Response: Thank you for the suggestion. “of G. biloba seeds” has been deleted. The sentence was rewritten as: “This was caused by the loss of water in the sarcotesta during the late stage of development”. (Line 203)
30. Line 193: These 3 values are related to the negative correlation between the embryo and the seed transverse diameter, water content and fresh weight. However, to avoid confusion (it may seem that these are p values!!), I highly suggest to add the p values in the round brackets. For example (transverse diameter correlation = -0.942, p =0.01; water content correlation = -0.908, p = 0.01; ecc...) See also my comment in the Table 2.
Response: The ambiguity might be caused by the unclear description in our manuscript, so we would like to say sorry to the reviewer. We have added p values in the round brackets to avoid confusion and we have revised Table 2.
The text has been modified as:
… (transverse diameter correlation = -0.942, p < 0.01; water content correlation = -0.908, p < 0.01; fresh weight correlation = 0.865, p < 0.01) … (Line 207-208)
31. Line 200: You should be coherent throughout the text. Replace" 140d-230d DAF" with "140-230 DAF"
Response: Thank you for the suggestion again, “140d-230d DAF” has been replaced with “140-230 DAF”. (Line 215)
32. Line 208: why the page number start from page 7? Please, authors should correct
Response: We would like to thank the reviewer for the comments about the page number. We have corrected it.
33. Line 208: Please, put it in italics
Response: Thank you for the suggestion again, the Latin/scientific names “Ginkgo biloba” have been corrected as italicized.
34. Line 209: Authors have to revise this table adding the P values. Here it is found the correlation between the embryo length and the other parameters, and, even if the * indicates the significant level of p in the correlation, the p values written in full are missing.
Response: Thank you for underlining this deficiency. According to the Reviewer’s comment, we have added p values to Table 2.
Table 2. Correlation analysis between embryo length, vertical diameter, transverse diameter, water content and fresh weight during seed development of Ginkgo biloba ‘Jiafozhi’
Indicators | Embryo Length | Transverse Diameter | Vertical Diameter | Water Content | Fresh Weight |
Embryo Length | 1 | -0.942** | -0.464* | -0.908** | -0.865** |
Sig. (2-tailed) | 0.000 | 0.015 | 0.000 | 0.000 | |
Transverse Diameter | 1 | 0.564** | 0.898** | 0.779** | |
Sig. (2-tailed) | 0.002 | 0.000 | 0.000 | ||
Vertical Diameter | 1 | 0.423* | 0.275 | ||
Sig. (2-tailed) | 0.028 | 0.165 | |||
Water Content | 1 | 0.809** | |||
Sig. (2-tailed) | 0.000 | ||||
Fresh Weight | 1 |
* Significant at 0.05 p level, ** Significant at 0.01 p level; others are non-significant.
35. Line 213: is it a mistake? maybe is it 88?
Response: Indeed, the 388 refers to the sum of the dropped seeds from September 8 (140 DAF) to October 28 (190 DAF).
36. Line 228: Authors mean 14d after sowing?
Response: Yes. What we wanted to express was “14d after sowing”. We have revised the sentence “On October 8 (170 DAF), the germination of excised embryos collected from the trees (T) and on the ground (G) of G. biloba seeds during 14d incubation were only 5.9% and 13.3%, respectively.” as “On October 8 (170 DAF), the germination percentages after a 14-day incubation of excised embryos of G. biloba seeds collected from trees (T) and from the ground (G) were only 5.9% and 13.3%, respectively”. (Line 242-243)
37. Line 229: What is S? Did you mean G? if yes, please replace S with G.
Response: Yes. We would like to say sorry to the reviewer and the editor for the careless mistake. Indeed, we misspelled “G” as “S”, and “S” has been replaced with “G” and we have checked the other parts of article.
38. Line 257: from sowing?
Response: We have revised “After 10d” as “At 10 days after sowing”. (Line 273)
39. Line 258: No, authors referred to figure 5. This mistake must be modified throughout the text
Response: Thank you for underlining this deficiency. Actually, we referred to figure 5. The mistakes have been modified throughout the text one by one. (Line 274-312)
“Figure 2A” in the Line 258 has been corrected as “Figure 5A”
“Figure 2B” in the Line 260 has been corrected as “Figure 5B”
“Figure 2C” in the Line 263 has been corrected as “Figure 5C”
“Figure 2C” in the Line 273 has been corrected as “Figure 5C”
“Figure 2D, E” in the Line 277 has been corrected as “Figure 5D, E”
“Figure 2F” in the Line 283 has been corrected as “Figure 5F”
“Figure 2I, J” in the Line 286 has been corrected as “Figure 5I, J”
“Figure 2K, L” in the Line 288 has been corrected as “Figure 5K, L”
“Figure 2M” in the Line 293 has been corrected as “Figure 5M”
40. Line 303: No, authors referred to figure 6. This mistake must be modified throughout the text.
Response: Thank you for underlining this deficiency. Actually, we referred to figure 6. The mistakes have been modified throughout the text one by one.
“Figure 5” in the Line 303 has been corrected as “Figure 6”
41. Line 307: No, authors authors referred to Table 5. This mistake must be modified
Response: Thank you for underlining this deficiency. We are very sorry for our incorrect writing, and we have been corrected these mistakes one by one in the revised manuscript.
“Table 3” in the Line 307 has been corrected as “Table 5”
42. Line 324: please, cite some works as example
Response: Thank you for the suggestion, we have cited some works as examples. The following text has been added:
… For example, Ball [23] confirmed maturation of tissues was not complete in the roots of these in vitro seedlings. Wang and Chen [24] suggested coconut milk, both filtered and autoclaved, was tested for its effect on the growth and structure of the young embryos …
The following reference has been added:
[23] Ball, E. Growth of the embryo of Ginkgo Biloba under experimental conditions. I. Origin of the first root of the seedling in vitro. American Journal of Botany 1956, 43, 488-495.
[24] Wang, F.H.; Chen, T.K. Experimental studies of young Ginkgo embryos—the effect of coconut milk on the embryos cultured in vitro. Acta Botanica Sinica 1965, 13, 224-231.
43. Line 347: No, is 230. Please, correct it
Response: “240 DAF” in the Line 347 has been corrected as “230 DAF”.
44. Line 352: Change with "Seed Germination"
Response: Thank you for the suggestion. We have replaced “Germination of Seed” with “Seed Germination”, The subtitle was rewritten as: “4.3. Seed Germination and Isolated Embryo”. (Line 376)
45. Line 356: I did not find this reference in the reference list. Please add this reference, unless it refers to [7] of the reference list. (see also my following comment). However, in [7] there is "Tsi-tung" is it a mistake? please verify
Response: Thank you for underlining this deficiency. We have added this reference according to the reviewer’s comment and removed the wrong reference number [7].
The following reference has been added:
[28] Li, T.T. The development of Ginkgo embryo in Vitro Sci. Rept. Nat. Tsing Hua Univ., Ser. B 1934, 7, 169-174.
46. Verify reference [28].
Response: We would like to thank the reviewer for the comments about references. The reference has been corrected to:
[38] Grobbelaar, N. Cycads, with Special Reference to the Southern African Species; Nat Grobbelaar: Pretoria, 2002. (Line 543)
47. Line 373: change this sentence in this way: the phase from unformed sperm to the proembryo could exist simultaneously in the same plant
Response: Thank you for the suggestion. The sentence “…even if the same plant, from the phase of unformed sperm to the proembryo in the same plant could exist simultaneously” has been revised as: “even in the same plant, because the phase from unformed sperm to the proembryo could exist simultaneously in the same plant” (Line 398-399)
48. Line 403-406: This paragraph has to be improved, revising English too. Try to change the text from line 403 to line 406 in order to avoid repeating the results.
Response: Thank you for the suggestion. We have revised this paragraph according to the reviewer’s comment. The sentence “While at 15℃, the germination process was very slow, and the initial time of seeds began to germinate was about 15 days later than that at 30 ℃. The germination percentage of seeds at 30 ℃ was significantly higher than that at 15 ℃ and 25 ℃. And the mean germination time (MGT) of seeds was significantly lower than that of 15℃ and 25℃.” has been revised as: “On the other hand, at 15 °C, the germination process was very slow. Although the germination percentage at 25 °C was also high, the MGT and SG had a better response at 30°C.” (Line 432-434)
49. In addition, the English must be revised, I highly recommended to submit the manuscript to a linguistic review process.
Response: We are grateful for the suggestion. We had sent the manuscript to MDPI English Editing Service for language service before submitting the revised version.
Thank you for suggestions, and all of your suggestions are very important for my thesis writing and research work. Thank you for your valuable advice again.
